# Survival Outcomes of Hepatectomy in Gastric Cancer Liver Metastasis: A Systematic Review and Meta-Analysis

**DOI:** 10.3390/jcm12020704

**Published:** 2023-01-16

**Authors:** Danny Conde Monroy, Milciades Ibañez-Pinilla, Juan Carlos Sabogal, Carlos Rey Chaves, Andrés Isaza-Restrepo, Felipe Girón, Marco Vanegas, Rafael Ibañez-Villalba, Lutz Mirow, Timo Siepmann

**Affiliations:** 1Department of Surgery, Hospital Universitario Mayor—Méderi, Bogota 110111, Colombia; 2Department of Surgery, Faculty of Medicine, Universidad del Rosario, Bogota 111221, Colombia; 3Division of Health Care Sciences, Dresden International University, 01067 Dresden, Germany; 4Faculty of Medicine, Fundación Universitaria Sanitas, Bogota 110131, Colombia; 5Department of Surgery, Faculty of Medicine, Pontificia Universidad Javeriana, Bogota 110231, Colombia; 6Department of Surgery, Klinikum Chemnitz gGmbH, Medical Campus Chemnitz, Medical Faculty Carl Gustav Carus, Technische Universität Dresden, 09116 Chemnitz, Germany; 7Department of Neurology, University Hospital Carl Gustav Carus, Technische Universität Dresden, 01307 Dresden, Germany

**Keywords:** gastric cancer, hepatectomy, meta-analysis, survival, mortality

## Abstract

Gastric cancer liver metastasis (GCLM) is a contraindication for surgical treatment in current guidelines. However, the results of recent studies are questioning this paradigm. We assessed survival outcomes and their predictors following hepatectomy for GCLM in a systematic review of studies published from 2000 to 2022 according to PRISMA guidelines. We identified 42,160 references in four databases. Of these, 55 articles providing data from 1990 patients fulfilled our criteria and were included. We performed a meta-analysis using random-effects models to assess overall survival (OS) and disease-free survival (DFS) at one, three, and five years post-surgery. We studied the impact of potential prognostic factors on survival outcomes via meta-regression. One, three, and five years after surgery, OS was 69.79%, 34.79%, and 24.68%, whereas DFS was 41.39%, 23.23%, and 20.18%, respectively. Metachronous presentation, well-to-moderate differentiation, small hepatic tumoral size, early nodal stage, R0 resection, unilobar compromisation, and solitary lesions were associated with higher overall survival. Metachronous presentation, smaller primary tumoral size, and solitary metastasis were linked to longer DFS. The results of our meta-analysis suggest that hepatectomy leads to favorable survival outcomes in patients with GCLM and provides data that might help select patients who will benefit most from surgical treatment.

## 1. Introduction

Gastric cancer (GC) is responsible for 1 in every 12 deaths globally. It represents approximately 8% of cancer-related deaths and is the third leading cause of cancer-related death worldwide [1,2]. In the surgical treatment of GC, complete resection, lymph node dissection, and neoadjuvant and/or adjuvant therapy are the goals for improved survival time [3,4,5,6]. However, despite advances in prevention and treatment strategies, the prognosis has only been modestly improved [2,3,4,5,6,7,8,9,10,11,12,13,14,15,16]. In Western countries such as the United Kingdom, the majority of gastric cancer cases are detected at an advanced stage. Only 30% of patients are eligible for treatment with curative intent, mainly due to metastatic spread [17,18,19,20,21,22,23,24,25]. The liver, bones, and peritoneum are the most common sites where metastases spread [21,22,23,24,25,26]. Gastric cancer liver metastases (GCLM) are often multifocal, bilobar, or accompanied by simultaneous extrahepatic metastases such as peritoneal lesions or extensive lymph node compromise [27]. The incidence of GCLM ranges from 5% to 9% [28]. They are synchronous (before, during, or within 6 months after gastrectomy) in 3% to 14% of patients and metachronous (6 months or later after gastrectomy) in up to 37% [29]. Current Western guidelines advise against surgical resection of liver metastases, with the majority of patients instead receiving palliative chemotherapy [25,26,27,28,29,30]. According to the National Comprehensive Cancer Network (NCCN), liver metastases are classified as IVb disease and are therefore recommended to be managed with systemic chemotherapy [31]. The median overall survival of GC patients with liver metastasis is around 12 months. The most widely used regimens are doublets or triplets with fluoropyrimidines, platinum derivatives, taxanes, or anthracyclines [32,33]. Given the unsatisfactory prognosis associated with systemic therapy, the clinical management of stage IV disease has been changing over the past few years. Advances in systemic therapy and liver surgery safety have driven the expansion of surgical indications for metastatic disease [34]. However, due to a lack of large, well-designed clinical trials, surgery remains limited to individually selected cases of GCLM. Solely in Japanese treatment guidelines, considering surgery with curative intent is recommended when the number of metastatic nodules is small and no other non-curable factors are present [35]. Here, we aimed to review the literature and provide meta-analytic evidence on survival outcomes following hepatectomy for GCLM. Furthermore, we sought to identify potential predictors of beneficial clinical outcomes.

## 2. Materials and Methods

We performed a systematic review and meta-analysis in compliance with the Preferred Reporting Items for Systematic Reviews and Meta-Analyses (PRISMA) guidelines. The PRISMA checklist is shown in the Appendix A.

### 2.1. Search Strategy

Two independent reviewers (D.C., C.R.) performed a systematic search of the literature to identify eligible articles. We searched the databases PubMed, Cochrane, EMBASE, and Google Scholar using the following search terms and Boolean operators: “Gastric cancer AND hepatectomy” OR “gastric AND cancer AND liver metastases” OR “gastric AND cancer AND metastasectomy” OR “stomach AND cancer AND liver metastases”, “liver resection”, “hepatectomy”, “carcinoma”, “neoplasm”. We also reviewed the reference list of included articles for eligible papers. The complete search strings for all databases are provided in Appendix A.

### 2.2. Study Selection and Data Collection

Publications were included when they met the following eligibility criteria: (i) studies involving humans and available as full-text articles in English published from January 2000 to June 2022; (ii) studies in patients who had hepatectomy for GCLM as an upfront radical resection with simultaneous management of the primary tumor and the liver metastasis, or as a subsequent procedure after primary gastric cancer treatment; (iii) articles reporting data on 1-,3-, and/or 5-year overall survival and/or disease-free survival of GC patients with only liver metastasis treated surgically; (iv) studies where planning of the surgery was based on the intention to achieve R0 resection. We included prospective or retrospective studies that were of observational or interventional nature. We excluded articles reporting on surgery for GCLM with insufficient reporting of outcome data as well as retrospective observations in less than 10 patients.

Two authors (D.C. and C.R.) independently selected studies by screening titles and abstracts and removed duplicates between articles identified across different databases. Subsequently, they interchanged the lists of articles deriving from their searches to compare the lists in a consensus-based approach. In the case of contradictions between the reviewers, a third reviewer (F.G.) was involved to establish consensus. Afterward, an investigator (D.C.) extracted relevant data from included articles and entered these data into a Microsoft Excel^®^ version 16.6 database.

### 2.3. Data Synthesis and Quality Assessment

Statistical analysis was performed using the Medcalc^®^ software package (Version 20.110, MedCalc Software Ltd., Acacialaan 22, 8400 Ostend, Belgium). A *p* value < 0.05 was considered to be statistically significant.

We calculated one-, three-, and five-year OS and DFS as the proportion of patients being alive and free from the tumor, respectively. Hazard ratio (HR) and estimated standard errors with corresponding 95% confidence intervals (95% CIs) were calculated to assess the association between potential prognostic factors and OS and DFS. HRs and their variance were obtained from the studies or calculated according to the data presentation: annual mortality rates, survival curves, number of deaths, or percentage freedom from death. A random-effects model was used to perform a meta-analysis assuming differences in the treatment effect. Forest plots were generated to illustrate the results of the meta-analysis. Publication bias was assessed with Egger and Begg tests and through funnel plots for graphical inspection. Heterogeneity across the studies was assessed using the Cochran Q test and/or the Higgins test (I^2^ statistic to measure the degree of variation not attributable to chance alone). Heterogeneity was graded as low (I^2^ < 25%), moderate (I^2^ = 25% to 75%), or high (I^2^ > 75%). We carried out a meta-regression including variables with a potentially prognostic association with survival outcomes. Models of meta-regression were built with one covariate at the time. We extracted data of sufficient quantity to conduct a meta-regression for age, N stage, time presentation, lobar compromise, number of lesions, size of the metastasis, and kind of hepatectomy for OS. By contrast, we were not able to perform a meta-regression for DFS due to the limited no availability of sufficient data. However, not enough data were obtained to perform a regression with all the factors. Additionally, a regression on the number of patients and year of publication was performed. A *p* value < 0.05 was considered to be statistically significant. Two raters (D.C. and C.R.) assessed study quality using the Newcastle–Ottawa quality assessment scale independently (36). Afterward, results were compared and a third rater (F.G.) was involved to achieve consensus where necessary.

## 3. Results

A total of 42,160 references were obtained by the initial electronic search: 9531 from PubMed, 264 from Cochrane, 22,480 from Google Scholar, and 9885 from Embase. Upon removal of duplicates, 183 papers were eligible for review of their abstracts and full texts. By applying our selection criteria, we included 55 suitable papers into our systematic review and meta-analysis of a total of 1990 patients who underwent hepatic resection for GCLM. All of the included studies were retrospective, and thus none of them had an interventional or prospective observational design. Table 1 and Table 2 summarize the main characteristics of the included studies. Figure 1 depicts the flow of information through the phases of our systematic review.

### 3.1. Characteristics of the Studies

All of the included articles report results from retrospective cohort studies. Study quality analysis using the Newcastle–Ottawa quality assessment showed that all of the included studies were of high quality and had a low risk of bias. Table 3 depicts the Newcastle–Ottawa classification.

Of the 55 included articles, 42 studies (76%) were conducted in Asian populations and 13 (24%) in Western countries. In all the studies involved, the indications for hepatectomy were good control of the primary tumor, no disseminated disease, and the feasibility of achieving R0 resection.

### 3.2. Characteristics of the Patients

The median age of our synthesized population was 64 years (range: 54–74 years), and the coefficient of variation was 7.1%. Men constituted 74.2% (n = 1385) and women 25.8% (n = 481) of the entire population. All patients had undergone surgery for primary GC either in a previous intervention or in the current intervention. All patients were treated with surgery for GCLM, either a synchronous resection in 1215 patients or a meta-chronous resection in 690 cases. The T classification was divided into two groups: the T1–T2 group consisted of 36.79% of the cases, and the T3–T4 group of 61.26%. The histology was classified according to the differentiation as well-differentiated (48.5%), moderately differentiated (38.1%), and poorly differentiated (33.9%). With respect to nodal compromise, 22.5% of patients were classified as N 0–1 and 38.5% as N 2–3. Of the entire population, 739 (37.13%) patients received adjuvant chemotherapy, and 159 (7.9%) received preoperative chemotherapy. Of all included articles (n = 55), 23 reported data on resection margins. In these studies, overall, 1070 cases had reported R0 resection margins, whereas they were R1 on pathologic analysis in 129 cases. Of the entire synthesized population, unilobar resection of metastasis was performed in 34.5% (n = 687) of patients, whereas multilobar resection was performed in 10.6% (n = 210). Solitary lesions were resected in 45.6% of cases (n = 908), whereas resections of multiple lesions were performed in 26.2% of cases (n = 522). The mean size of the hepatic lesions resected was 3.12 cm (range: 2–5.5 cm). A minor hepatectomy was performed in 54.37% (n = 1082) of the population. By contrast, a major hepatectomy was undertaken in only 18.49% (n = 368) of patients. The mean follow-up in the included studies was 34.6 months (range: 8.9–90.8 months).

### 3.3. Mortality at 30 Days

The mortality rate at 30 days was 1.37% in the synthesized population. The data reported were insufficient to describe the operative complication subtypes and co-morbidities associated with complications.

### 3.4. Survival Outcomes

#### 3.4.1. Overall Survival at 1 Year

Of the 55 included articles, 45 (81.8%) reported OS in the first year after hepatectomy for GCLM. The pooled OS rate was 69.8% (CI 95%: 65.5%, 73.8%) with moderate heterogeneity and random effect size (I^2^ = 69.50%, heterogeneity test Q, *p* < 0.0001). There was no evidence of publication bias, neither on Begg (*p* = 0.9057) and Egger (*p* = 0.9057) tests nor on inspection of the funnel plot (Figure 2).

#### 3.4.2. Overall Survival at 3 Years

Of the 55 included articles, 44 (80%) reported OS in the third year after hepatectomy for GCLM. The pooled OS rate was 34.8% (CI 95%: 30.7%, 38.9%) with a random effect size of and moderate heterogeneity (I^2^ = 65.7%, heterogeneity test Q, *p* < 0.0001). There was no evidence of publication bias, neither on Begg (*p* = 0.85) or Egger (*p* = 0.90) tests nor on the inspection of the funnel plot (Figure 3).

#### 3.4.3. Overall Survival at 5 Years

Of the 55 included articles, 48 (87.2%) reported OS in the fifth year after hepatectomy for GCLM. The pooled OS rate was 24.7% (CI 95%: 21.3%, 28.1%) with random effect size and moderate heterogeneity (I^2^ = 62.4%, heterogeneity test Q, *p* < 0.0001). There was no evidence of publication bias, neither on Begg (*p* = 0.0997) and Egger (*p* = 0.2975) tests nor on inspection of the funnel plot (Figure 4).

#### 3.4.4. Disease-Free Survival at 1 Year

Of the 55 included articles, 16 (29%) reported DFS in the fifth year after hepatectomy for GCLM. The pooled DFS rate was 41.4% (CI 95%: 34.4%, 48.5%) with a random effect size and moderate heterogeneity (I^2^ = 69.5%, heterogeneity test Q, *p* < 0.0001). There was no evidence of publication bias, neither on Begg (*p* = 0.4421) and Egger (*p* = 0.5197) tests nor on inspection of the funnel plot (Figure 5).

#### 3.4.5. Disease-Free Survival at 3 Years

Of the 55 included articles, 14 (25.4%) reported DFS in the third year after hepatectomy for GCLM. The pooled DFS rate was 23.2% (CI 95%: 18.2%, 28.7%) with random effect size and moderate heterogeneity (I^2^ = 59%, heterogeneity test Q, *p* < 0.0001). There was no evidence of publication bias, neither on Begg (*p* = 0.2259) and Egger (*p* = 0.0893) tests nor on inspection of the funnel plot (Figure 6).

#### 3.4.6. Disease-Free Survival at 5 Years

Of the 55 included articles, 16 (29%) reported DFS in the fifth year after hepatectomy for GCLM. The pooled DFS rate was 20.2% (CI 95%: 14.3%, 26.7%), with random effect size and moderate heterogeneity (I^2^ = 74.6%, heterogeneity test Q, *p* < 0.0001). There was no evidence of publication bias, neither on Begg (*p* = 0.8571) and Egger (*p* = 0.5429) tests nor on inspection of the funnel plot (Figure 7).

### 3.5. Prognostic Factors

In the meta-analysis of the risk factors, we found an association between improved OS and the following factors: R0 resection, small diameter of the metastasis, resection of a solitary lesion, unilobar localization, low node compromise, early T stage of the primary tumor, well-to-moderate differentiation grade, and metachronous presentation. In relation to DFS, the factors associated with improved DFS were a metachronous presentation, solitary lesions, and an early T stage (Table 4).

### 3.6. Meta-Regression

The results of the meta-regression are shown in Table 5. Multivariate models were not constructed due to a large amount of missing data. Among all variables tested, only synchronous presentation displayed a prognostic association with higher OS (Table 5).

## 4. Discussion

The major finding of this systematic review and meta-analysis is that hepatectomy results in beneficial rates of both overall survival and disease-free survival when assessed one, three, and five years post-surgery. The survival outcomes observed at these time points were 69.79%, 34.79%, and 24.68% for OS and 41.39%, 23.23%, and 20.18% for DFS, respectively, with a median survival of 24.5 months. Periprocedural mortality was 1.37%. Our results are in line with previous meta-analyses. The analysis of Petrelli et al. showed a weighted median OS of 22 months at 5 years in 23 studies with 870 patients taken to hepatectomy for GCLM [37]. Another meta-analysis from 2016 assessed 39 studies; the authors described a median survival rate of 68% at 1 year, 31% at 3 years, and 27% at 5 years in GCLM patients [38]. These survival outcomes are consistent with our analysis and differ substantially from those reported with systemic therapy using epirubicin, oxaliplatin, and capecitabine (EOX) in the REAL3 randomized controlled phase III trial at 1 year of 46%, and with a median survival of 11.3 months [39].

Metastatic gastric cancer has long been considered an aggressive disease, and therefore not suitable for surgery [32]. Current guidelines do not support surgery for GCLM [31]. Moreover, there is widespread skepticism about performing surgery in these cases [40]. A survey applied to surgeons in Europe and Japan found that for metachronous GCLM, most of the specialists (50.4%) prefer preoperative chemotherapy followed by liver resection, whereas 30.3% preferred chemotherapy alone and 36% preferred alternative treatments such as ablative radiofrequency ablation (RFA) alone or RFA with chemotherapy [40]. Some of the reasoning for performing surgery on GCLM patients is inspired by the results of research on colorectal LM. In 1439 patients with hepatectomy for colorectal liver metastasis, Adam et al. described survival rates of 33% and 23% at 5 and 10 years, respectively [41]. Recently, a multicenter retrospective review was conducted on 144 patients who underwent hepatectomy in synchronous and metachronous settings. They identified a median OS of 12 months [42]. Historically, OS has been the most commonly used metric for judging the success of treatment [43]. The main disadvantage of this measure is the need for extended follow-up and the potentially diluted death measurement due to other nonmalignant causes. DFS has emerged as a potential candidate for a surrogate of OS in various malignant diseases [43,44,45]. It may complement OS in the measurement of survival outcomes. However, reports of DFS in the literature are scarce. Only 16 out of the 55 articles included in our review reported DFS. A review of studies on a total of 1573 patients who underwent hepatic resection described 1-, 3-, and 5-year DFS of 44%, 24%, and 22%, respectively [46]. Survival is dependent on multiple recognized demographic, tumoral, and metastatic prognostic factors (see Table 3) [43]. Among the primary tumor characteristics, histologic type, serosal invasion, N-stage, and metachronous presentation are prognostic factors (*p* < 0.05) [32,47]. Tiberio et al. [48] recently demonstrated that T stage, R0 resection, and adjuvant chemotherapy administration are independent factors of OS. T stage ≥ 3, high nodal compromise (N2–3), and poor differentiation were identified as negative prognostic factors for both synchronous and metachronous metastases in their study. They conclude that these patients should be carefully evaluated before hepatic resection is proposed. In this meta-analysis, not achieving an R0 resection margin was associated with the worst prognosis (HR4.04 IC:2.73–7.08, *p* < 0.001). Even in the face of the worst prognosis of bilobar compromise and the size of the lesion, a propensity score analysis of 119 patients who received multidisciplinary treatments for liver metastasis showed that in the presence of an R0 resection, the distribution and number of liver metastases do not affect the prognosis [49]. Our synthesized analysis highlights the prognostic value of solitary resection over multiple lesions. However, liver resection is not clearly limited to a specific number of metastases. Some reports showed the benefit of resecting 1–3 metastases, and even multiple lesions cannot be considered exclusion criteria for surgery [50].

Neoadjuvant and adjuvant therapies are fundamental in treating advanced GC patients. Any progression during chemotherapy is probably the most relevant contraindication for surgery [46]. Naturally, the use of systemic therapy has improved in recent years, and it seems appropriate to discuss the role of preoperative chemotherapy to increase survival. The FLOT3 trial evaluated the benefit of a regimen of FLOT (5-fluorouracil, oxaliplatin, and docetaxel) followed by surgery and confirmed the potential OS gained [51]. In our review, neoadjuvant chemotherapy administration data were insufficient for inclusion in our analyses. Preoperative chemotherapy was less frequently administered than postoperative chemotherapy in the cohorts we analyzed. In our review, only four studies reported hazard ratios for adjuvant chemotherapy.

Currently, no prospective data exist about the comparison between non-resectional management and surgical management in patients with GCLM. Most of the studies in this review have no systemic therapy arm [52]. Shinohara et al. retrospectively compared OS in 22 patients receiving surgery in GCLM to 25 patients who did not receive surgery and found a significant difference between the groups (median survival time: 22 vs. 7 months, respectively, *p* = 0.001) [53]. Despite the benefits of gastrectomy plus hepatectomy over non-resectional management in patients with GCLM, it must be pointed out that all data came from retrospective studies and systematic reviews.

The use of targeted therapies has increased given the advances in the understanding of the molecular mechanisms of GC. Currently, the most studied therapy is related to HER2 expression. While early studies disagreed on the prognostic relevance of HER status, recent evidence highlights its value [54,55]. In 2010, the Toga Study evaluated the use of the HER2-targeted monoclonal antibody trastuzumab with standard chemotherapy in 584 patients. The addition of trastuzumab increased the median survival in HER2 positive patients to 13.8 months compared to 11.1 months with chemotherapy alone (HR 0.74 (95% CI 0.60–0.91, *p* = 0.0046)) [56]. More recent trials support the use of targeted therapy in HER2 patients (Keynote 811 and Destiny Gastric 01) [57]. However, the therapy has some drawbacks: HER2 expression in gastric cancer is only around 9–38%, the antibody shows high heterogeneity, the benefit is more evident in patients with high levels of HER2 expression, and there are different testing methods for HER2 measurement (2). Moreover, in GCLM, the evidence is limited. From the studies evaluated, only 1 study discriminated HER2 patients and received trastuzumab associated with chemotherapy [58]. A study performed in 94 patients with GCLM found no relevance of HER2 positivity as an independent prognostic factor (HR 0.918, IC: 0.185–4.5), but the analysis seems to be affected by the underpowered sample [59]. Future studies should include this as a potential prognostic factor.

The relevant difference between synchronous and metachronous diseases may be related to the insufficient data in the literature on metachronous cases. This may be explained by the difficulty in finding patients with potential surgical indications, tumoral aggressiveness, or, in most cases, the simultaneous spread of metastases. In a 2017 retrospective cohort study, the authors compared the outcomes of 653 patients with metachronous disease; 34 were treated surgically, while 619 were treated non-surgically. In this study, surgically treated patients displayed higher survival outcomes than non-surgically treated patients (1YOS: 73.5% vs. 19.7%, 3YOS: 36.9% vs. 6.6%, 5YOS: 24.53% vs. 4.4%, *p* < 0.001) [60]. Cui et al. found the metachronous hepatectomy to be a favorable factor for OS, but the number of liver metastases was not [61]. Overall, a metachronous presentation was associated with improved OS in this study. On the other hand, in this study’s meta-regression, synchronous presentation was associated with better OS in the first year. In addition to survival differences between metachronous and synchronous presentations, Tatsubayashi et al. described postoperative complications as more common in patients with synchronous GCLM compared to metachronous disease. The length of hospital stay in these patients was also prolonged (*p* = 0.003) [62].

This work contributes to the growing evidence supporting hepatectomy for GCLM. It has strengths. Our systematic review and meta-analysis provide synthesized analyses of a large population of patients with GCLM, complementing and partially exceeding the scope of previous reviews. It supports the value of DFS as a complementary marker of survival in GC patients. However, our analysis is limited by the retrospective nature of the included studies. Thus, selection bias and institutional bias cannot be ruled out.

## 5. Conclusions

In conclusion, the results of our systematic review and meta-analysis in a large population of patients with GCLM indicate that hepatectomy leads to favorable survival outcomes in these patients. While our analysis provides data that might help select patients who will benefit most from surgical treatment, large, well-designed prospective studies are needed to confirm these observations. Based on our analysis and the current literature, we advocate using DFS as a complementary survival outcome parameter in research on GC.

## Figures and Tables

**Figure 1 jcm-12-00704-f001:**
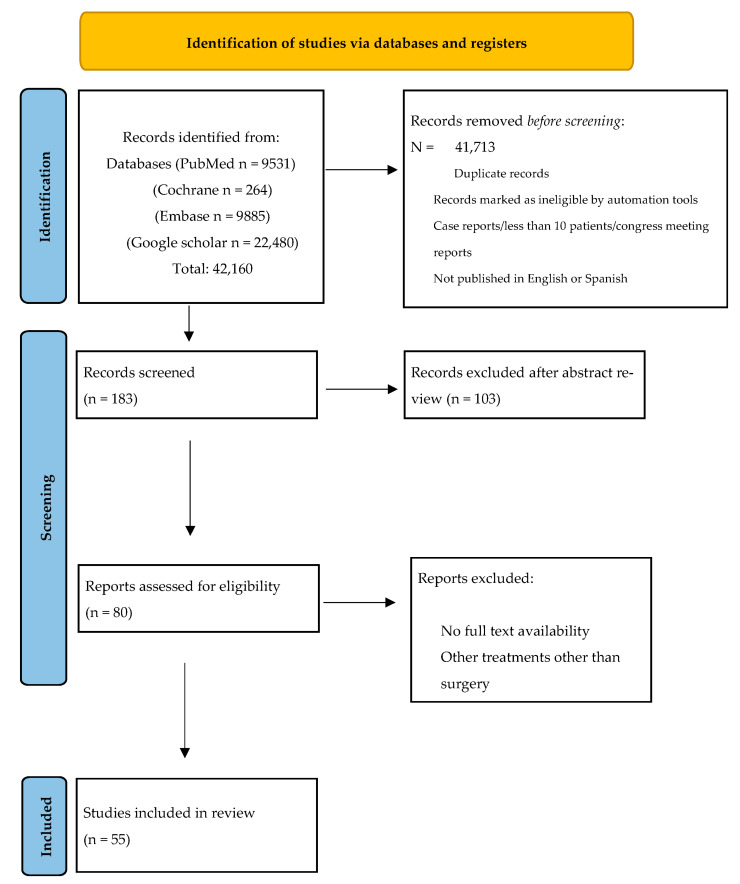
Flow diagram depicting the flow of information through the different phases of our systematic review. PRISMA, Preferred Reporting Items for Systematic Reviews and Meta-Analyses.

**Figure 2 jcm-12-00704-f002:**
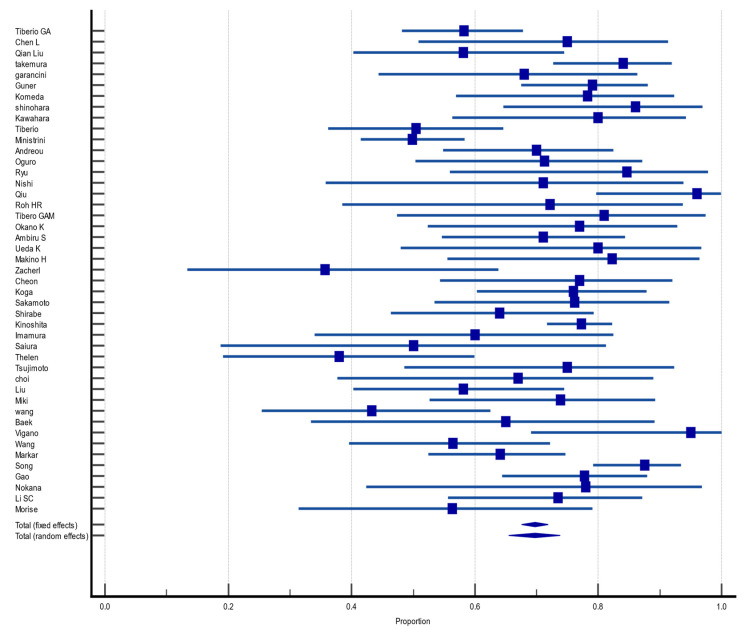
Forest plot of overall survival at 1 year after hepatectomy for gastric cancer liver metastasis (GCLM). Description of the proportion and confidence intervals at 95% in the studies that reported 1-year overall survival.

**Figure 3 jcm-12-00704-f003:**
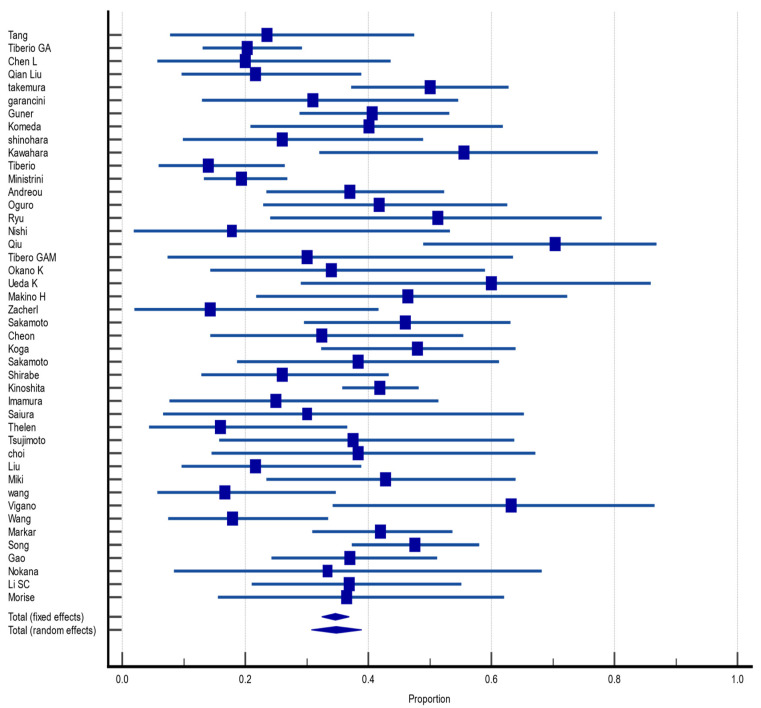
Forest plot of overall survival at 3 years after hepatectomy for gastric cancer liver metastasis (GCLM). Description of the proportion and confidence intervals at 95% in the studies that reported 3-year overall survival.

**Figure 4 jcm-12-00704-f004:**
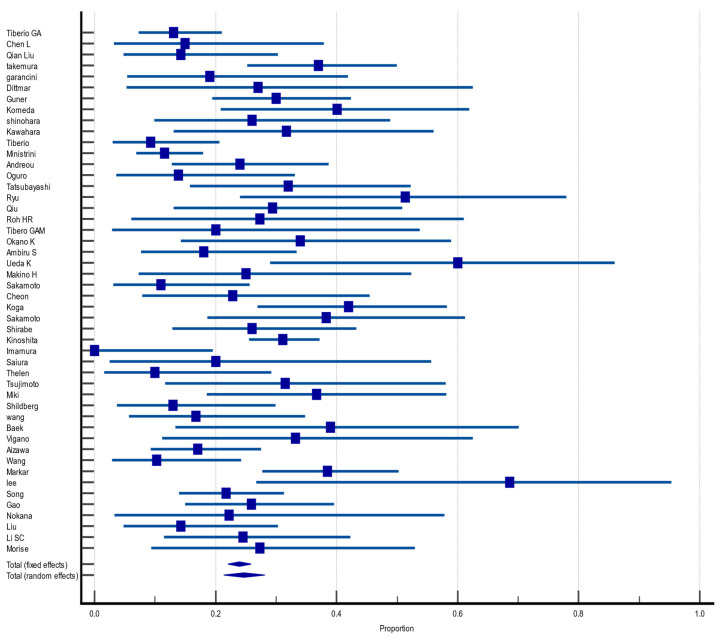
Forest plot of overall survival at 5 years after hepatectomy for gastric cancer liver metastasis (GCLM). Description of the proportion and confidence intervals at 95% in the studies that reported 5-year overall survival.

**Figure 5 jcm-12-00704-f005:**
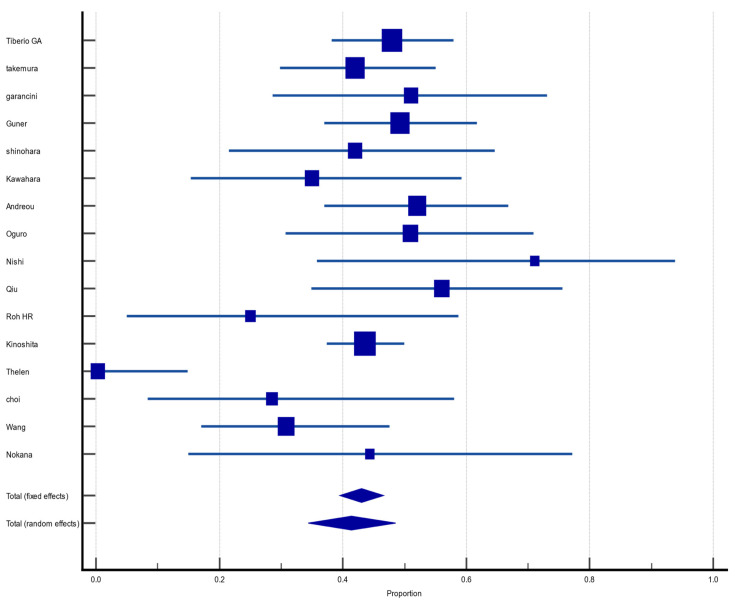
Forest plot of disease-free survival at 1 year after hepatectomy for gastric cancer liver metastasis (GCLM). Description of the proportion and confidence intervals at 95% in the studies that reported 1-year disease-free survival.

**Figure 6 jcm-12-00704-f006:**
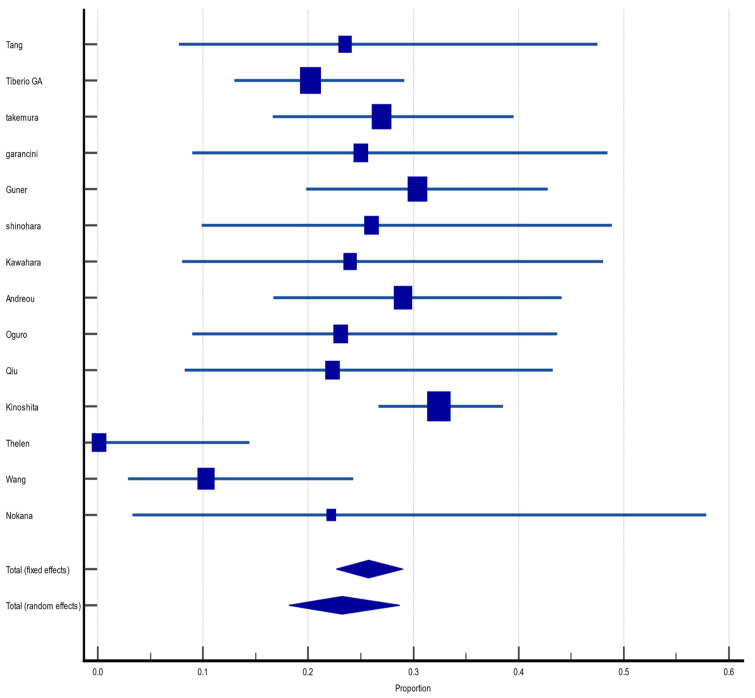
Forest plot of disease-free survival at 3 years after hepatectomy for gastric cancer liver metastasis (GCLM). Description of the proportion and confidence intervals at 95% in the studies that reported 3-year disease-free survival.

**Figure 7 jcm-12-00704-f007:**
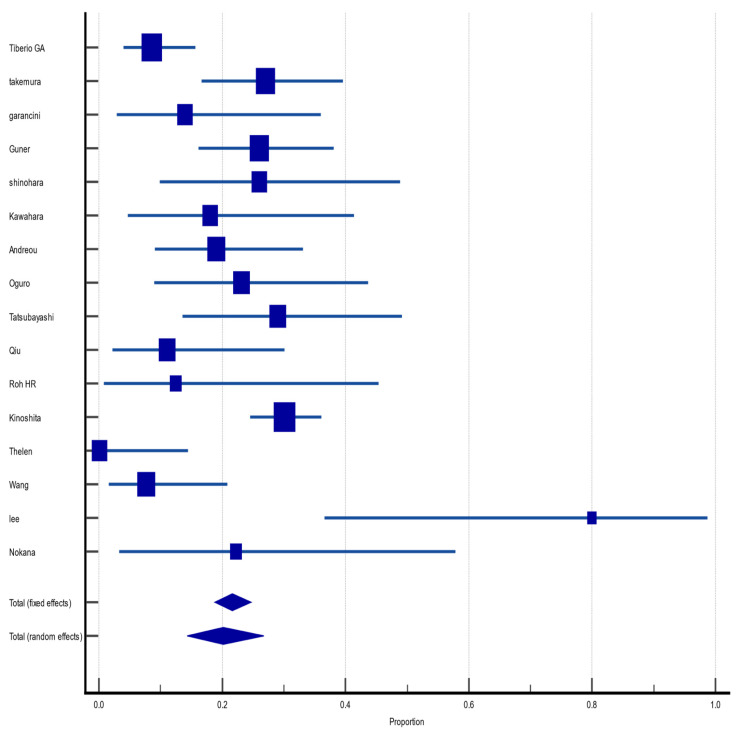
Forest plot of disease-free survival at 5 years after hepatectomy for gastric cancer liver metastasis (GCLM). Description of the proportion and confidence intervals at 95% in the studies that reported 5-year disease-free survival.

**Table 1 jcm-12-00704-t001:** Study characteristics.

Study	Year	N	Median Age	Synchronous (n)	Metachronous (n)	Ethnicity (n)	Male (n)	Female (n)	N 0–1 (n)	N 2–3 (n)	T1–T2 (n)	T3–T4 (n)	Histology: Well Differentiated (n)	Histology: Moderately Differentiated (n)	Histology: Poorly Differentiated (n)	Adj Chemo (n)	Neoadj Chemo (n)
Tang	2020	20	61	19	1	East	16	4	10	10	2	18	0	0	12	17	0
Kawahara	2020	20	73.5	11	9	East	13	7	8	12	NR	4	3	11	3	20	0
Gao	2019	54	57	NR	NR	East	43	11	18	36	29	25	NR	NR	NR	24	NR
Nokana	2019	10	68	4	6	East	9	1	7	3	3	7	NR	NR	NR	0	0
Nishi	2018	10	71.7	6	4	East	9	1	NR	NR	8	12	NR	NR	NR	8	3
Ministrini	2018	144	68	112	32	West	94	50	48	68	23	93	13	NR	22	32	20
Shirashu	2018	9	74	6	3	East	8	1	NR	NR	NR	NR	9	NR	0	3	NR
Ryu	2017	14	NR	NR	NR	East	0	0	NR	NR	NR	NR	NR	NR	NR	NR	NR
Lee	2017	7	59.2	NR	NR	East	5	2	NR	NR	NR	NR	NR	NR	NR	6	0
Song	2017	96	63	59	37	East	72	24	28	68	47	59	62	NR	34	58	0
Li SC	2017	34	62	NR	34	East	23	11	NR	NR	NR	NR	NR	NR	NR	NR	NR
Oguro	2016	26	69.5	6	20	West	23	3	NR	8	8	18	8	10	8	15	NR
Tatsubayashi	2016	28	72	15	13	East	23	5	3	25	8	20	22	0	6	12	3
Markar	2016	78	65	78	0	West	14	51	NR	NR	NR	NR	NR	NR	NR	NR	NR
Guner	2016	68	61	26	42	West	56	12	32	36	17	52	45	NR	23	66	0
Tiberio GA	2016	105	68	74	31	West	71	34	36	40	38	46	NR	NR	NR	29	NR
Shinohara	2015	22	66.7	13	9	East	19	3	NR	NR	NR	3	18	NR	4	16	6
Tiberio	2015	52	68	52	0	West	0	0	NR	NR	NR	NR	NR	NR	NR	22	0
Kinoshita	2015	256	64	106	150	East	207	49	54	204	74	NR	173	NR	NR	84	45
Liu	2015	35	56	35	0	West	22	13	NR	NR	6	29	14	21	0	0	0
Ohkura	2015	13	63	9	4	East	13	0	NR	NR	NR	NR	NR	NR	NR	12	0
Qian Liu	2015	35	56	35	NR	East	22	13	4	31	6	29	14	NR	20	35	0
Komeda	2014	24	69.5	1	23	East	21	3	10	14	17	7	NR	NR	NR	15	11
Andreou	2014	47	62	34	13	East	32	15	NR	NR	NR	NR	NR	NR	NR	16	16
Aizawa	2014	74	66	74	0	East	56	18	NR	NR	NR	NR	NR	NR	NR	NR	NR
Wang	2014	39	64	39	0	East	26	13	23	16	8	31	23	NR	16	39	0
Qiu	2013	25	NR	25	0	East	22	3	4	21	17	8	9	NR	16	14	4
Chen L	2013	20	54	20	NR	East	12	8	12	8	6	14	2	14	4	20	20
Baek	2013	12	61	3	9	East	11	1	9	3	3	9	9	NR	1	6	NR
Vigano	2013	14	61.5	9	5	West	0	0	NR	NR	NR	NR	NR	NR	NR	0	8
Takemura	2012	64	65	34	30	East	49	15	22	42	NR	49	42	NR	22	26	18
Garancini	2012	21	64	12	9	West	14	7	19	11	NR	8	8	NR	13	NR	NR
Dittmar	2012	10	57	NR	NR	West	0	0	NR	NR	NR	NR	NR	NR	NR	NR	0
Liu	2012	35	NR	35	0	East	29	8	12	23	19	16	NR	NR	25	NR	NR
Miki	2012	25	72	16	9	East	23	2	14	11	8	17	NR	NR	NR	10	0
Shildberg	2012	31	65	17	14	West	20	11	NR	NR	NR	NR	NR	NR	NR	9	2
Wang	2012	30	60	30	0	East	27	3	10	20	4	26	NR	NR	NR	30	0
Tsujimoto	2010	17	66	9	8	East	16	1	12	5	12	5	NR	NR	NR	14	0
Choi	2010	14	65	NR	14	East	11	3	NR	NR	NR	NR	NR	NR	NR	10	NR
Makino H	2010	16	NR	9	7	East	13	3	NR	NR	NR	NR	10	NR	6	11	3
Tibero GAM	2009	11	NR	NR	11	West	0	0	NR	NR	NR	NR	NR	NR	NR	0	0
Ueda K	2009	12	67	12	0	East	0	0	NR	NR	27	45	NR	NR	NR	8	NR
Thelen	2008	24	64	15	9	East	17	7	0	NR	8	16	1	7	16	NR	NR
Cheon	2008	22	59	18	4	East	18	4	NR	NR	NR	NR	NR	9	12	NR	NR
Morise	2008	18	64	11	7	East	16	2	NR	NR	NR	NR	NR	NR	NR	NR	NR
Koga	2007	42	64	20	22	East	30	12	20	21	NR	14	NR	18	13	13	0
Sakamoto	2007	37	64	16	21	East	29	8	NR	NR	25	12	NR	NR	NR	6	0
Roh HR	2005	11	61.3	8	3	East	10	1	7	NR	1	10	2	4	1	NR	NR
Sakamoto	2003	22	63	12	10	East	13	9	NR	NR	NR	NR	NR	NR	NR	8	NR
Shirabe	2003	36	66	16	20	East	33	3	NR	NR	NR	NR	NR	NR	NR	NR	NR
Zacherl	2002	15	61.6	10	5	West	10	5	NR	NR	5	4	NR	NR	NR	NR	NR
Okano K	2002	19	69	13	6	East	13	6	16	NR	11	8	6	NR	13	6	NR
Saiura	2002	10	60.5	6	4	East	7	3	NR	NR	NR	NR	NR	NR	NR	6	0
Imamura	2001	17	65	7	10	East	15	2	NR	NR	NR	NR	5	7	5	NR	NR
Ambiru S	2000	40	63	18	22	East	30	10	10	30	28	12	NR	NR	NR	13	0

NR = not reported. N = number of patients analyzed in the study. Synchronous = number of patients with synchronous liver metastases. Ethnicity = “West” for studies conducted in Western countries and “East” for studies conducted in Asian countries. Male = number of male patients. N 0–1,2–3 = number of patients with lymph-node involvement of primary cancer in the four different stages (0–3). T = number of patients with stage T1–T4. Histology defines the differentiation grade of the primary tumor. Adjuvant chemo and Neoadj chemo = number of patients who were administered adjuvant or neoadjuvant chemotherapy.

**Table 2 jcm-12-00704-t002:** Metastasis characteristics and survival outcomes.

Study	Year	R0 (n)	Unilobar (n)	Multilobar (n)	Solitary (n)	Multiple (n)	Diameter (cm)	Size of the Lesion < 3 (n)	Minor Hepatectomy (n)	Major Hepatectomy (n)	30-Day Mortality (n)	1 DFS (%)	3 DFS (%)	5 DFS (%)	Median DFS (Months)	Median Follow-up (Months)	1 YOS (%)	3 YOS (%)	5 YOS (%)	Median Survival (Months)
Tang	2020	NR	17	3	16	4	2.9	NR	NR	NR	3	NR	23.5	NR	NR	NR	NR	23.5	NR	20
Hara	2020	NR	NR	NR	11	9	2.5	14	NR	NR	0	35	24	18	10.5	77	80	55.5	31.7	52
Gao	2019	NR	NR	NR	38	16	NR	NR	NR	NR	NR	NR	NR	NR	NR	NR	77.8	37	25.9	29.3
Nokana	2019	NR	NR	NR	7	3	NR	NR	NR	NR	NR	44.4	22.2	22.2	NR	NR	78	33.3	22.2	30
Nishi	2018	10	NR	NR	6	4	2.3	5	5	5	0	71.1	NR	NR	40	12.4	71.1	17.8	NR	24.5
Ministrini	2018	117	NR	NR	NR	NR	NR	NR	132	12	3	NR	NR	NR	NR	NR	49.9	19.4	11.6	12
Shirashu	2018	9	5	4	0	9	2.5	NR	1	8	0	NR	NR	NR	7.9	47.9	NR	NR	NR	24.8
Ryu	2017	NR	NR	NR	NR	NR	4.2	NR	7	7	0	NR	NR	NR	NR	NR	84.6	51.3	51.3	NR
Lee	2017	NR	6	1	5	2	NR	NR	NR	NR	NR	NR	NR	80	74.1	NR	NR	NR	68.6	67.5
Song	2017	91	57	29	42	54	NR	NR	61	35	0	NR	NR	NR	NR	33	87.5	47.6	21.7	34
Li SC	2017	NR	NR	NR	NR	NR	NR	NR	NR	NR	NR	NR	NR	NR	NR	NR	73.5	36.9	24.5	26.1
Oguro	2016	NR	NR	NR	16	10	3.7	NR	NR	NR	NR	50.9	23.1	23.1	43	77	71.3	41.8	13.9	25
Tatsubayashi	2016	28	20	8	20	8	2.4	NR	27	1	0	NR	NR	29	47	26	NR	NR	32	49
Markar	2016	NR	NR	NR	NR	NR	NR	NR	66	12	10	NR	NR	NR	NR	NR	64.1	42	38.5	NR
Guner	2016	NR	60	8	45	23	2.7	60	47	21	1	49.3	30.4	26	NR	NR	79.1	40.6	30	24
Tiberio GA	2016	89	NR	NR	NR	NR	NR	NR	94	11	1	48	20.2	8.6	10	NR	58.2	20.3	13.1	14.6
Shinohara	2015	NR	17	5	11	11	NR	NR	NR	NR	0	42	26	26	22	NR	86	26	26	22
Tiberio	2015	52	NR	NR	NR	NR	NR	NR	38	14	0	NR	NR	NR	NR	NR	50.4	14	9.3	13
Kinoshita	2015	230	NR	NR	168	88	3	NR	183	73	2	43.6	32.4	30.1	9.4	65	77.3	41.9	31.1	31.3
Liu	2015	32	30	5	27	8	NR	24	29	6	0	NR	NR	NR	NR	40	NR	NR	14.3	33
Ohkura	2015	NR	NR	NR	4	9	NR	NR	NR	NR	0	NR	NR	NR	NR	NR	NR	NR	NR	NR
Qian Liu	2015	32	30	5	27	8	NR	NR	29	6	0	NR	NR	NR	NR	41	58.1	21.7	14.3	33
Komeda	2014	NR	NR	NR	17	7	NR	NR	11	13	0	NR	NR	NR	NR	NR	78.3	40.1	40.1	22.3
Andreou	2014	41	33	14	NR	NR	2	NR	34	13	0	52	29	19	NR	71	70	37	24	18
Aizawa	2014	53	NR	NR	31	22	NR	NR	NR	NR	0	NR	NR	NR	NR	90.8	NR	NR	17	13
Wang	2014	39	34	5	31	8	2.8	NR	NR	NR	0	30.8	10.3	7.7	8	13.9	56.4	17.9	10.3	14
Qiu	2013	NR	21	4	19	6	2	13	19	6	0	56	22.3	11.1	18	42	96	70.4	29.4	38
Chen L	2013	NR	11	9	8	12	4.1	NR	6	14	0	NR	NR	NR	NR	9.9	75	20	15	22.3
Baek	2013	11	11	1	11	1	NR	NR	9	3	0	NR	NR	NR	NR	12.5	65	NR	39	31
Vigano	2013	NR	18	2	9	5	NR	NR	8	6	0	NR	NR	NR	NR	42.5	95	63.2	33.2	52.3
Takemura	2012	55	NR	NR	37	27	NR	NR	50	14	0	42	27	27	9	27	84	50	37	
Garancini	2012	19	16	5	12	9	3	NR	17	4	0	51	25	14	NR	21.6	68	31	19	
Dittmar	2012	NR	NR	NR	NR	NR	2.6	NR	8	2	0	NR	NR	NR	NR	40.3	NR	NR	27	
Liu	2012	NR	12	23	12	23	NR	NR	NR	NR	NR	NR	NR	NR	NR	38	58.1	21.7	NR	15
Miki	2012	NR	20	5	18	7	2	NR	NR	NR	NR	NR	NR	NR	5	NR	73.9	42.8	36.7	33.4
Shildberg	2012	23	30	1	NR	NR	NR	NR	21	10	2	NR	NR	NR	NR	NR	NR	NR	13	NR
Wang	2012	NR	27	3	22	8	3.7	NR	23	7	0	NR	NR	NR	NR	11	43.3	16.7	16.7	11
Tsujimoto	2010	17	13	5	13	4	4.8	NR	11	6	0	NR	NR	NR	NR	29.3	75	37.5	31.5	34
Choi	2010	NR	11	3	9	5	NR	NR	8	6	NR	28.5	NR	NR	NR	15.25	67	38.3	NR	NR
Makino H	2010	NR	11	5	9	7	NR	8	NR	NR	NR	NR	NR	NR	NR	15.9	82.3	46.4	25	38
Tibero GAM	2009	NR	NR	NR	NR	NR	NR	NR	NR	NR	NR	NR	NR	NR	NR	19	81	30	20	23
Ueda K	2009	NR	NR	NR	NR	NR	NR	NR	8	NR	NR	NR	NR	NR	NR	8.9	80	60	60	NR
Thelen	2008	17	18	6	18	6	5.5	NR	16	8	1	33	10	10	NR	18	38	16	10	18
Cheon	2008	22	21	1	18	4	2.4	NR	NR	NR	1	NR	NR	NR	NR	15.5	77	32.4	22.8	17
Morise	2008	NR	15	3	14	4	NR	NR	14	4	NR	NR	NR	NR	NR	NR	56.3	36.5	27.3	13
Koga	2007	36	NR	NR	29	13	NR	20	NR	NR	NR	NR	NR	NR	NR	16	76	48	42	34
Sakamoto	2007	32	30	7	21	16	3.8	NR	32	5	0	NR	NR	NR	NR	NR	NR	46	11	31
Roh HR	2005	NR	11	0	11	0	NR	NR	10	1	0	25	NR	12.5	8	19	72.2	NR	27.3	19
Sakamoto	2003	NR	17	5	16	6	3	NR	19	3	0	NR	NR	NR	NR	17	76.2	38.3	38.3	21.4
Shirabe	2003	NR	NR	NR	31	5	NR	NR	NR	NR	0	NR	NR	NR	NR	NR	64	26	26	NR
Zacherl	2002	NR	10	4	8	6	NR	7	2	13	0	NR	NR	NR	NR	51	35.7	14.3	NR	15.7
Okano K	2002	NR	12	7	10	9	3.9	NR	16	NR	0	NR	NR	NR	NR	36	77	34	34	21
Saiura	2002	NR	7	3	4	6	NR	NR	NR	NR	3	NR	NR	NR	NR	NR	50	30	20	25
Imamura	2001	15	12	5	8	9	NR	NR	NR	NR	NR	NR	NR	NR	NR	24	60	25	0	NR
Ambiru S	2000	NR	24	16	19	21	NR	NR	21	19	0	NR	NR	NR	NR	88	71.1	NR	18	12

NR = not reported. R0 = number of patients who achieved an R0 surgical removal on both primary cancer and liver metastases. Unilobar and multilobar = number of patients with unilobar or multilobar liver involvement. Solitary or multiple = number of patients with solitary or multiple liver metastases. Diameter =diameter in cm of liver metastasis. Size of the lesion = divided in size < or > to 3 cm. major or minor hepatectomy = defined as the number of patients with hepatectomy of 4 or more segments (major) or <4 segments (minor). 30-day mortality = number of patients dead after 30 days of surgery. 1 DFS = percentage of patient’s disease free at 1 year. 3 DFS = percentage of patient’s disease free at 3 years. 5 DFS = percentage of patient’s disease free at 5 years. median DFS = months of disease-free survival. Median F.U. = median follow-up time in years. 1 YOS = percentage of patients surviving at 1 year. 3 YOS = percentage of patients surviving at 3 years. 5 YOS = percentage of patients surviving at 5 years. median survival = median survival in months.

**Table 3 jcm-12-00704-t003:** Study quality rating via the Newcastle–Ottawa scale [36].

Study	Year	Quality of the Study Selection	Comparability	Outcome	Quality Score NOS Final
Tang	2020	4	2	3	9
Kawahara	2020	4	2	2	8
Gao	2019	3	2	3	8
Nokana	2019	4	2	3	9
Nishi	2018	3	2	3	8
Ministrini	2018	4	2	3	9
Shirashu	2018	4	2	3	9
Ryu	2017	4	2	3	9
Lee	2017	4	2	3	9
Song	2017	4	2	3	9
Li SC	2017	4	2	3	9
Oguro	2016	3	2	2	7
Tatsubayashi	2016	4	2	3	9
Markar	2016	3	2	3	8
Guner	2016	4	2	3	9
Tiberio GA	2016	3	2	3	8
Shinohara	2015	4	2	3	9
Tiberio	2015	4	2	2	8
Kinoshita	2015	4	2	3	9
Liu	2015	4	2	3	9
Ohkura	2015	4	2	3	9
Qian Liu	2015	4	2	2	8
Komeda	2014	4	2	2	8
Andreou	2014	4	2	2	8
Aizawa	2014	3	2	3	8
Wang	2014	3	2	3	8
Qiu	2013	4	2	3	9
Chen L	2013	4	2	3	9
Baek	2013	3	2	3	8
Vigano	2013	3	2	3	8
Takemura	2012	4	2	2	8
Garancini	2012	4	2	2	8
Dittmar	2012	4	2	2	8
Liu	2012	4	2	3	9
Miki	2012	4	2	3	9
Shildberg	2012	3	2	3	8
Wang	2012	3	2	3	8
Tsujimoto	2010	4	2	3	9
Choi	2010	3	2	3	8
Makino H	2010	4	2	3	9
Tibero GAM	2009	4	2	3	9
Ueda K	2009	4	2	3	9
Thelen	2008	3	2	3	8
Cheon	2008	3	2	3	8
Morise	2008	3	2	3	8
Koga	2007	4	2	3	9
Sakamoto	2007	4	2	3	9
Roh HR	2005	3	2	3	8
Sakamoto	2003	4	2	3	9
Shirabe	2003	4	2	3	9
Zacherl	2002	4	2	3	9
Okano K	2002	4	2	3	9
Saiura	2002	4	2	2	8
Imamura	2001	3	2	3	7
Ambiru S	2000	4	2	3	9

The Newcastle–Ottawa Scale quality rating tool evaluates the selection, comparability, and outcome of a study with a possible maximum score of 9 points. Studies scoring 7–9 points are considered high quality, 4–6 points high risk of bias, and 0–3 points very high risk of bias.

**Table 4 jcm-12-00704-t004:** Prognostic associations with survival outcomes.

	Overall Survival			Disease-Free Survival		
Prognostic Factor	Hazard Ratio	Heterogeneity	*p* Value	Hazard Ratio	Heterogeneity	*p* Value
Age	1.06 (0.79–1.42)	64.93%	0.666	1.39 (0.78–2.45)	76.70%	0.254
Sex	0.99 (0.55–1.7)	0%	0.998			
Metachronous presentation	1.48 (1.15–1.89)	55%	0.002	1.50 (1.21–1.86)	0%	<0.001
Well–moderate differentiation grade	1.34 (1.13–1.59)	0%	0.001	1.27 (0.80–2.01)	0%	0.310
Early T stage (T1–T2)	1.66 (1.25–2.21)	0%	<0.001	2.21 (1.19–4.08)	41.57%	0.011
Low node compromise (0–1)	1.60 (1.32–1.95)	21.06%	<0.001			
Unilobar localization	1.71 (1.01–2.89)	10.7%	0.046			
Solitary lesions	1.57 (1.12–2.18)	59.28%	0.008	2.34 (1.67–3.29)	0%	<0.001
Small diameter (<3 cm)	1.26 (1.06–1.50)	49.34%	0.008	1.81 (0.92–3.56)	36.44%	0.083
Adjuvant chemotherapy	1.66 (0.73–3.76)	0%	0.221			
R0 resection	4.04 (2.73–7.08)	0%	<0.001			

Prognostic factors’ hazard ratios on overall survival and disease-free survival, the degree of heterogeneity, and *p* value.

**Table 5 jcm-12-00704-t005:** Meta-regression of prognostic associations.

Variable	Number of Studies	Coefficient	*p* Value	R^2^	Test for Residual Heterogeneity (I^2^ = %)
Year of publication	45	0.0246	0.213	0.00	67.29
Number of included patients	45	0.0002	0.921	0.00	66.83
Age	39	0.0063	0.811	0.00	67.70
N stage 2–3	23	0.6492	0.424	0.10	71.74
Synchronous presentation	42	−0.8050	0.018	0.33	57.87
Unilobar presentation	27	0.5977	0.524	0.00	63.44
Solitary lesions	36	−0.4672	0.553	0.00	59.47
Multiple lesions	36	0.4597	0.554	0.00	58.43
Size > 3 cm	7	1.4394	0.540	0.00	65.45
Minor hepatectomy	29	−0.2090	0.791	0.06	73.36

Meta-regression based on a random effect size model. R^2^ identifies the heterogeneity accounted for by the model.

## Data Availability

The datasets used and/or analyzed during the current study are available from the corresponding author upon reasonable request.

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
