# Peer review of "Survival Outcomes of Hepatectomy in Gastric Cancer Liver Metastasis: A Systematic Review and Meta-Analysis"

_jcm, 2023, doi:10.3390/jcm12020704_

Round 1

Reviewer 1 Report

I agree that resection of liver metastases contributes to the prognostic association. However, the results of chemotherapy in patients with liver metastases who are considered operable are not clear. If you have the  studies of  chemotherapy for liver limited metastasis, please present them.

Reviewer 2 Report

Overall, the study of this meta-analysis is very meaningful, providing us with some management strategies for gastric cancer liver metastases and informing us of the major prognostic factors in patients with gastric cancer liver metastases. The design of this paper is reasonable, the meta-analysis and statistical methods used are accurate, and the conclusions are reliable.

However, the author needs to clarify that for the treatment of gastric cancer patients with liver metastasis, if metachronous resection of liver metastases is performed, some patients will use targeted therapy drugs. The author did not mention this treatment method in the paper, nor did he show this part of data in the literature. However, targeted therapy is also very important for gastric cancer liver metastasis, which will affect patients' OS.

Reviewer 3 Report

Comments:

In this manuscript entitled “Survival Outcomes of Hepatectomy in Gastric Cancer Liver Metastasis. A Systematic Review and Meta-Analysis” the author(s) have properly reviewed the relevant literature from 2000 to 2022 to assess the survival outcomes following hepatectomy for GCLM. They have included a significant number of literature providing data from a significant number of patients as well. They also performed a meta-analysis using random effects model to study OS and DFS at 3 different time points post-surgery.

The idea is well conceived and planned with a good search strategy, data collection and synthesis, and quality assessment. The results are nicely presented with ample supporting information, flow diagram, tables, and figures. The findings of the study that hepatectomy is beneficial for OS and DFS post-surgery are quite significant in terms of clinical settings.

I would recommend for the acceptance of the article in its current form after the correction of some minor errors/typos:

1.      Line 22, correct “overall (OS)” to “overall survival (OS)”.

2.      Line 76, correct " articles to for eligible” to “articles for eligible”.

3.      Line 191-192, correct “performed in in 34.5%” to “performed in 34.5%”.

4.      Line 285, correct "Our results are in line previous meta-analyses” to “Our results are in line with the previous meta-analyses”.

Reviewer 4 Report

Dear author,

1-                The submitted manuscript Survival Outcomes of Hepatectomy in Gastric Cancer Liver Metastasis. A Systematic Review and Meta-Analysis presents a good study for review the literature and provide meta-analytic evidence on survival outcomes following hepatectomy for GCLM. 

2-                The introduction part is explanatory and covers the specific knowledge about the topic.

3-                Check punctuation marks, word spacing and English throughout the introduction.

4-                Please write full words of all abbreviations for first mention.

5-                Check punctuation marks, word spacing and English throughout the results and discussion (line 192, 267).

6-                Please rearrange the table better.

7-                Use a standardized format for references writing throughout the manuscript.

8-                The limitations of the study should be given in the Discussion section.
